# Compartmentalized *oskar* degradation in the germ plasm safeguards germline development

**Catherine E Eichler[†], Anna C Hakes, Brooke Hull, Elizabeth R Gavis***

Department of Molecular Biology, Princeton University, Princeton, United States

**Abstract** Partitioning of mRNAs into ribonucleoprotein (RNP) granules supports diverse regulatory programs within the crowded cytoplasm. At least two types of RNP granules populate the germ plasm, a cytoplasmic domain at the posterior of the *Drosophila* oocyte and embryo. Germ granules deliver mRNAs required for germline development to pole cells, the germ cell progenitors. A second type of RNP granule, here named founder granules, contains *oskar* mRNA, which encodes the germ plasm organizer. Whereas *oskar* mRNA is essential for germ plasm assembly during oogenesis, we show that it is toxic to pole cells. Founder granules mediate compartmentalized degradation of *oskar* during embryogenesis to minimize its inheritance by pole cells. Degradation of *oskar* in founder granules is temporally and mechanistically distinct from degradation of *oskar* and other mRNAs during the maternal-to-zygotic transition. Our results show how compartmentalization in RNP granules differentially controls fates of mRNAs localized within the same cytoplasmic domain.

**\*For correspondence:**
gavis@princeton.edu

[†]CEE has previously published as Catherine E Ruesch

**Competing interests:** The authors declare that no competing interests exist.

## Introduction

The earliest stages of animal development generally occur in the absence of transcription. Thus, spatial and temporal control of gene activity required during this crucial period must be imposed post-transcriptionally. Due in part to its relatively late zygotic genome activation (ZGA) and consequent reliance on maternally synthesized transcripts, *Drosophila* has served as a valuable model for studying post-transcriptional gene regulation. Mechanisms including mRNA localization, translational control, and mRNA degradation are critical for establishing the body axes, for germline development, and for the elimination of maternal transcripts during the maternal-to-zygotic transition (MZT) when maternal products are replaced by zygotic gene products.

Post-transcriptional control can be coordinated through the partitioning of mRNAs into ribonucleoprotein (RNP) granules. These non-membrane bound bodies serve to compartmentalize regulatory events within the cytoplasm, such as sequestration of translationally repressed mRNAs in stress granules and P bodies (*Guzikowski et al., 2019*). RNP granules can also promote colocalization and co-regulation of mRNAs that function in the same biological process and the segregation of different mRNAs with conflicting regulatory needs (*Buchan, 2014*). Such roles in colocalization and segregation are exemplified by granules in the *Drosophila* germ plasm.

Hundreds of mRNAs are highly enriched in the germ plasm, a specialized cytoplasm that forms at the posterior of the *Drosophila* oocyte and persists in the early embryo, where it induces the formation of the germ cell progenitors, called pole cells, and specifies germline fate (*Jambor et al., 2015*; *Lécuyer et al., 2007*; *Mahowald, 2001*). Many of these mRNAs are co-packaged in RNP assemblies called germ granules, which coordinate their segregation to the transcriptionally quiescent pole cells (*Lerit and Gavis, 2011*; *Little et al., 2015*; *Trcek et al., 2015*). Among these are mRNAs like *nanos* that function in germline development (*Kobayashi et al., 1996*). Not all germ plasm enriched mRNAs are partitioned into germ granules, however. One such transcript, *oskar*, encodes a core

germ granule protein whose synthesis at the posterior of the oocyte initiates germ plasm assembly. Production of Oskar is targeted to the posterior of the oocyte by transport of *oskar* mRNA to the posterior, where its translation is activated. Oskar protein then nucleates formation of germ granules with the recruitment of additional core granule proteins (*Lehmann, 2016*). As oogenesis progresses, *nanos* and numerous other transcripts populate the germ granules (*Forrest and Gavis, 2003*; *Little et al., 2015*). By contrast, *oskar* mRNA is not incorporated into germ granules but forms large RNP bodies, together with the double-stranded RNA-binding protein Staufen, that contain several to hundreds of *oskar* transcripts (*Little et al., 2015*). Because germ plasm formation begins with *oskar* mRNA localization (*Lehmann, 2016*), we refer to the *oskar*-containing RNPs in the germ plasm as founder granules. Notably, although both germ granule destined mRNAs and *oskar* are highly enriched in the germ plasm by virtue of their respective granule association, the localized transcripts represent only a small fraction of their total masses in the embryo (≤4% and 18%, respectively) (*Bergsten and Gavis, 1999*; *Little et al., 2015*; *Trcek et al., 2015*). The majority remain distributed throughout the cytoplasm, where they are silenced by additional layers of post-transcriptional regulation and ultimately degraded during the MZT.

The *Drosophila* embryo develops as a syncytium in which nuclei undergo 13 rounds of semi-synchronous divisions (nuclear cycles; nc) before the major wave of ZGA and cellularization occur. The pole cells form precociously, however, during nc10 when cell membrane buds enclose nuclei that have entered the germ plasm (*Figure 1—figure supplement 1*). Whereas both germ granules and founder granules persist in the germ plasm after fertilization, only the germ granules become enriched in the pole cells. This distinction is important, as targeting *oskar* to germ granules and consequent accumulation of *oskar* in pole cells impairs germline development, decreasing the number of pole cells formed and disrupting their migration to the gonad during gastrulation (*Little et al., 2015*). We now report that *oskar* mRNA itself, and not its capacity to produce Oskar protein, is toxic to pole cells. Such deleterious effects caused by mis-incorporation of *oskar* in pole cells highlight the need to sequester *oskar* from germ granule mRNAs within the germ plasm and to prevent its entry into pole cells.

Here, we investigate how incorporation of *oskar* mRNA into pole cells is minimized, and the role of founder granules in this process. We show that, in contrast to germ granule mRNAs, *oskar* is degraded in the germ plasm beginning in nc7. This selectivity occurs through the recruitment of mRNA decay factors specifically by founder granules. The subsequent degradation of *oskar* results in founder granule destabilization and ultimately limits uptake of *oskar* by pole cells. Founder granules impart another level of spatiotemporal control by promoting *oskar* degradation in the germ plasm well in advance of the degradation of unlocalized *oskar* during the MZT. Finally, we demonstrate that degradation of *oskar* in the germ plasm occurs by a process distinct from the degradation of unlocalized *oskar* during the MZT and requires the Piwi protein Aubergine. Together, these results show how the fates of different mRNAs with similar localizations can be controlled by their compartmentalization into distinctive RNPs.

## Results

### Incorporation of untranslatable *oskar* mRNA into pole cells impairs germline development

The adverse effects on pole cell development caused by forcing *oskar* into germ granules (*Little et al., 2015*) could be due to *oskar* mRNA itself or to an excess of Oskar protein in the pole cells. To distinguish between these possibilities, we targeted an untranslatable *oskar* transcript to pole cells. *oskar* mRNA is translated to produce two isoforms, Long Oskar and Short Oskar, using two alternative start codons. We engineered mutations that change both start codons to arginines (M1, 139R) in the *oskΔi1,2-nos3′UTR* (*oΔn*) transgene, which expresses *oskar* mRNA carrying the *nanos* 3′UTR for germ granule targeting instead of its own localization elements (*Figure 1—figure supplement 2A*). The *oΔn* and *o^{M1,139R}Δn* transcripts were expressed at comparable levels (*Figure 1—figure supplement 1B*), but only *oΔn* produced Oskar protein in addition to the endogenous Oskar present in the embryos (*Figure 1—figure supplement 2C,D*). Similarly to *oΔn* mRNA, *o^{M1,139R}Δn* colocalized with germ granules and was incorporated into pole cells (*Figure 1A–F*).

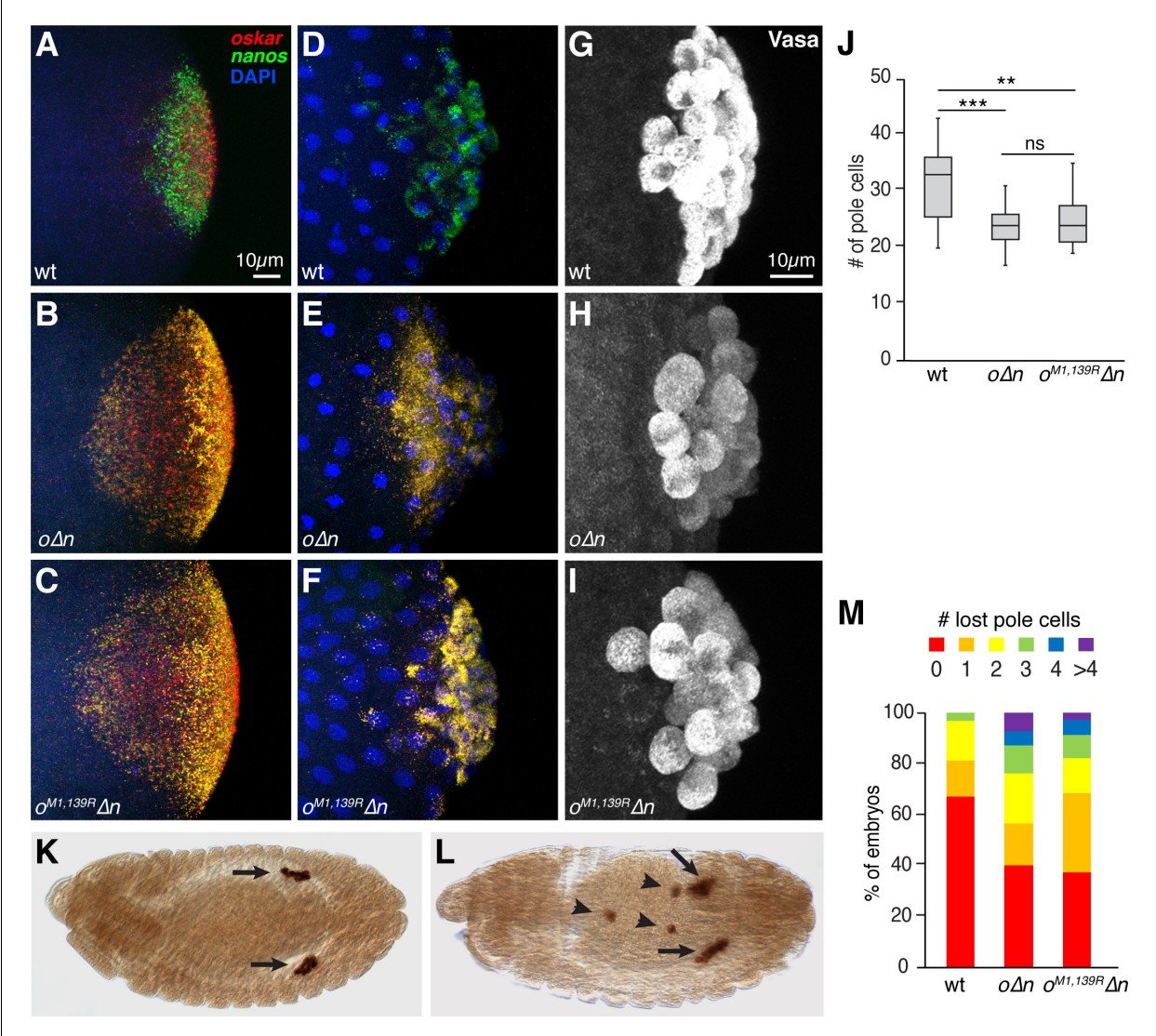

**Figure 1.** Incorporation of *oskar* mRNA into pole cells hinders their development independent of translation. (A–F) Confocal z-series projections of the posterior region of wild-type (wt) embryos (A,D), *oΔn* embryos (B, E) and *o^{M1,139R}Δn* (C,F). *oskar* (red) and *nanos* (green) mRNAs were detected by FISH in embryos prior to (A–C) and after (D–F) pole cell formation. Both endogenous *oskar* (in founder granules) and the transgenic *oskar* transcripts (co-localized with *nanos* in germ granules) are detected in (B,C,E,F). (G–I) Confocal z-series projections of pole cells, marked by anti-Vasa immunofluorescence, in wt (G), *oΔn* (H), and *o^{M1,139R}Δn* (I) embryos. (J) Quantification of the number of pole cells formed in genotypes shown in (G–I), n = 15 embryos each. Box and whisker plot shows median, upper and lower quartiles, and range. (K, L) Anti-Vasa immunostaining of stage 13 *o^{M1,139R}Δn* embryos showing examples of proper pole cell migration (K) and defective migration (L). Arrows indicated pole cells coalescing into gonads. Arrowheads indicate 'lost' pole cells that fail to reach the gonad. (M) Quantification of the frequency of embryos with the indicated numbers of lost pole cells observed in late-stage embryos; n = 91–136 embryos each. **p<0.01, ***p<0.001 as determined by Student's t-test. In this and all subsequent figures, embryos are oriented with the posterior end to the right, dorsal side toward the top.

The online version of this article includes the following source data and figure supplement(s) for figure 1:

**Source data 1.** Related to *Figure 1J and M*.
**Figure supplement 1.** Overview of *Drosophila* nuclear cycles and MZT.
**Figure supplement 2.** Quantification of transgene mRNA and protein levels.
**Figure supplement 2—source data 1.** Related to *Figure 1—figure supplement 2B and D*.

Like embryos from females expressing $o\Delta n$, embryos from females expressing $o^{M1,139R}\Delta n$ formed fewer pole cells than wild-type embryos (*Figure 1G–J*). The $o^{M1,139R}\Delta n$ embryos also showed pole cell migration defects similar to those of $o\Delta n$ embryos, with many pole cells failing to reach the gonad (*Figure 1K–M*). Thus, mis-incorporation of *oskar* mRNA itself in germ granules interferes with germline development, independently of Oskar protein production. The toxicity of *oskar* mRNA could result from titration of one or more factors involved in germline development, similarly to the proposed noncoding function of *oskar* during oogenesis (*Jenny et al., 2006*; *Kanke et al., 2015*). These data strongly suggest the requirement for a mechanism to regulate *oskar* at the mRNA level in the embryonic germ plasm to prevent its potential threat to germline development.

## Founder granules have limited motility and do not disperse during pole cell formation

In wild-type embryos, only low levels of *oskar* are detected in pole cells (*Figure 2A*). Our results therefore suggest that segregation of *oskar* from germ granule mRNAs by packaging into founder granules prevents enrichment of *oskar* in pole cells, where it impairs their development. We next investigated how this regulation is accomplished. Germ granules become enriched in the pole cells by virtue of their accumulation at centrosomes associated with posterior nuclei. This accumulation occurs by dynein-mediated transport on astral microtubules and triggers the budding of the plasma membrane around the nuclei and associated germ granules to form the pole cells (*Lerit and Gavis, 2011*). During mid-oogenesis, *oskar* RNPs are transported to the posterior of the oocyte by kinesin (*Brendza et al., 2000*; *Gáspár et al., 2017*). Thus, in contrast to the behavior of germ granules, founder granules might be dispersed by kinesin-mediated transport away from the pole cell nuclei, preventing their accumulation in pole cells.

To address this possibility, we compared the behavior of founder granules marked with Staufen-GFP and germ granules marked with Vasa-mCherry in live embryos. As previously observed (*Lerit and Gavis, 2011*), germ granules exhibited directed motility toward embryonic nuclei at the onset of pole cell formation. By contrast, most founder granules appeared to jiggle in place (*Video 1*). Tracking of both germ granules and founder granules showed that founder granules moved more slowly, with smaller displacements, and with fewer linear trajectories than germ granules (*Figure 2B–D*). Whereas germ granules became clustered around nuclei as a result of their directed motility, the less motile founder granules remained uniformly distributed in the germ plasm (*Figure 2E–I*). Consequently, founder granules were detected among the germ granules clustered around nuclei (*Figure 2F,H*) and near the cortex (*Figure 2H,I*), suggesting that they can be passively incorporated in pole cells. Moreover, since founder granules do not disperse from the vicinity of the forming pole cells, uptake of *oskar* by pole cells must be limited by another mechanism.

## Germ plasm-localized *oskar* is degraded during pole cell formation

In our live imaging experiments, we observed an apparent decrease in the number of founder granules as pole cells formed, suggesting they were destabilized. To further investigate this behavior, we visualized founder granule and germ granule constituent mRNAs, *oskar* and *nanos* respectively, before, during, and after pole cell formation by quantitative fluorescence in situ hybridization (FISH). As measured by total fluorescence intensity, the amount of *oskar* mRNA in the germ plasm decreased by 98% from before pole cell formation to after pole cell formation (*Figure 3A,C*). Consistent with previous studies showing that germ granule mRNAs are maintained in the germ plasm throughout pole cell formation (*Bashirullah et al., 2001*), there was no significant change in the amount of *nanos* (*Figure 3B,C*). We also compared the behavior of *oskar* targeted to germ granules with that of *oskar* in founder granules. To discriminate germ granule associated *oskar* from endogenous *oskar* in the same embryos, we used an *oskΔi-sfgp-nos3'UTR* transgene (*oΔgn*), in which *oskar* is tagged with superfolder *gfp* sequences and compared it to a control *osk-sfgfp* transgene (*og*). FISH analysis showed that the amount of germ plasm localized *og* mRNA decreased by 90% from before to after pole cell formation, whereas *oΔgn* mRNA remained largely unchanged (*Figure 3—figure supplement 1*). These results are consistent with the specific degradation of founder granule associated *oskar* mRNA during pole cell formation.

mRNA degradation typically involves removal of the 5' cap followed by exonuclease digestion. In *Drosophila*, decapping is carried out by Decapping Protein 2 (DCP2) and its cofactor Decapping

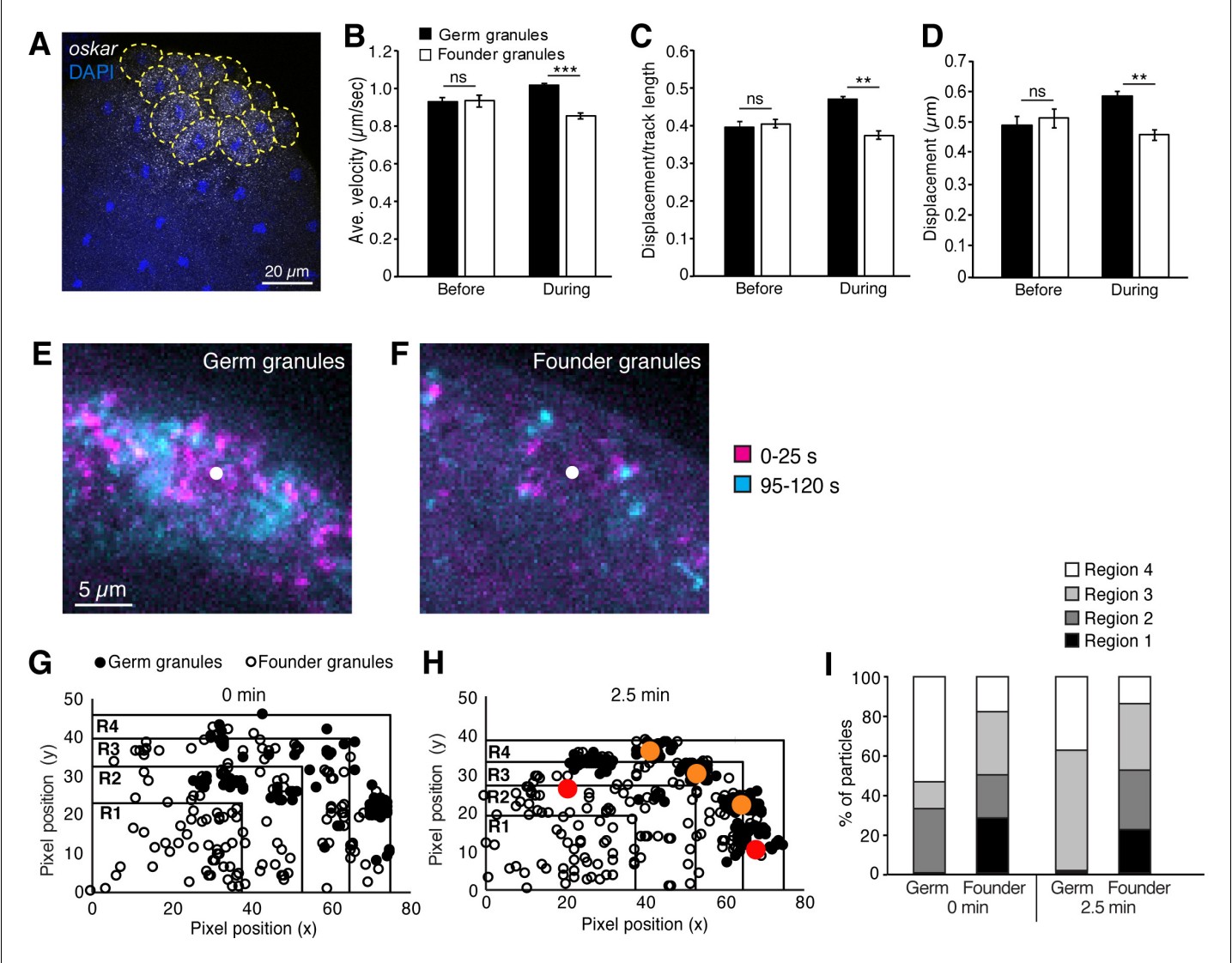

**Figure 2.** Founder granules are less motile than germ granules during pole cell formation. (A) Confocal z-series projection showing *oskar* mRNA in an embryo at nc10. Yellow dashed lines outline pole cells. (B–D) Quantification of germ granule and founder granule motility prior to and during pole cell formation: average velocity (B); linearity (C); and displacement (D). For each time point, three 2 min movies, 491–3221 tracks each, were analyzed. (E, F) Maximum projections of the first 25 s (magenta) and last 25 s (cyan) of a 2 min movie taken during pole cell formation showing the positions of germ granules (E) and founder granules (F) at the start and end of each movie. The white dot indicates the center of the nucleus. Images are cropped to show one pole cell each from *Video 1*. (G–I) Quantification of the distribution of germ granules and founder granules before and during pole cell formation. Plots of the locations of germ granules (filled circles) and founder granules (open circles) as the onset of (G) and 2.5 min into pole cell formation (H). Red and orange dots indicate approximate positions of nuclei in and out of the frame, respectively. Plots are divided into 4 regions of equal area with region four closest to the cortex and region one farthest from the cortex. The number of each type of granule within each region is quantified in (I); n = 108–248 particles. Values are mean ± s.e.m.; \*\*p<0.01; \*\*\*p<0.001 as determined by Student's t-test.

The online version of this article includes the following source data for figure 2:

**Source data 1.** Related to *Figure 2B*.
**Source data 2.** Related to *Figure 2C*.
**Source data 3.** Related to *Figure 2D*.
**Source data 4.** Related to *Figure 2G–I*.

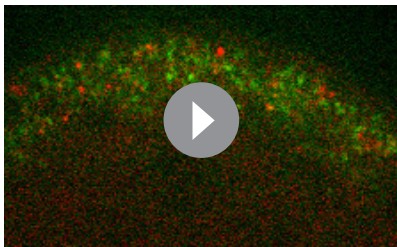

**Video 1.** Behavior of germ granules and founder granules prior to pole cell formation. Time-lapse movie of germ granules labeled with Vasa-mCherry (green) and founder granules labeled with GFP-Staufen (red). The movie starts as nuclei have reached the posterior of the embryo and induce release of granules from the cortex. Embryo is oriented with the posterior toward the top of the frame. Images were captured at five frames/second for a total of 600 frames and the movie is shown at 100 frames/second.

https://elifesciences.org/articles/49988#video1

Protein 1 (DCP1) (*Siwaszek et al., 2014*). To confirm that *oskar* is degraded in the germ plasm, we visualized DCP1 by immunofluorescence together with *oskar* by FISH in embryos from nc2 to nc12. Within the germ plasm, the frequency of detecting a particle containing one or more *oskar* transcripts colocalized with DCP1 began to increase at nc5 and continued to increase through pole cell formation (*Figure 4A–C* and data not shown). Consistent with the differential stability of *oskar* and germ granule mRNAs in the germ plasm, we did not detect colocalization of DCP1 with a representative germ granule mRNA, *cyclin B* (*Figure 4—figure supplement 1*). In addition, we found that 77% and 49% of *oskar* particles that colocalized with DCP1 also colocalized with the decapping complex associated protein Me31B (*Figure 4D,D',F*) and the 5' to 3' exonuclease Pacman (*Figure 4E,E',G*), respectively. Temporal analysis showed that recruitment of Pacman was delayed relative to DCP1 (*Figure 4C*), indicating that the degradation machinery is recruited in a sequential manner. Notably, DCP1 did not colocalize with *oΔgn* mRNA in germ granules (*Figure 4H–I*), suggesting that the degradation machinery is specifically recruited to founder granule associated *oskar*. We conclude that *oskar* degradation dramatically reduces the amount of *oskar* mRNA available for incorporation into pole cells.

## *oskar* mRNA is degraded within founder granules

We next considered what accounts for the specific vulnerability of *oskar* in the germ plasm. In addition to *oskar* mRNA levels, the sizes of founder granules, as measured by the number of *oskar* mRNAs they contain (see Materials and methods), decreased over the course of pole cell formation (*Figure 3D*). By contrast, germ granule size distributions, measured by the number of *nanos* transcripts they contain, were unchanged (*Figure 3E*). Two scenarios are consistent with the observed decrease in founder granule size. In one, founder granules disassemble, leaving *oskar* accessible to the degradation machinery. In contrast, other mRNAs that remain packaged in RNP granules, like those in germ granules, would remain protected from the degradation machinery in the germ plasm. Alternatively, founder granules could serve as the site of *oskar* degradation, thereby compartmentalizing the activity of the degradation machinery to preserve germ granule mRNAs.

To distinguish whether founder granule disassembly occurs before or after *oskar* degradation, we compared the timing of: 1) *oskar* association with degradation factors; 2) changes in total *oskar* levels; and 3) changes in founder granule composition from nc2 to nc12. As described above, colocalization of *oskar* with DCP1 first began to increase at nc5 (*Figure 4C*). The amount of *oskar* in the germ plasm decreased subsequently at nc7 (*Figure 5A*), and the average size of *oskar* particles declined sharply at nc9, when pole cells form (*Figure 5B*). The amount of Staufen in founder granules, measured by immunofluorescence intensity, began to decrease at nc11 (*Figure 5C*) along with a dramatic decrease in the percentage of Staufen particles that remained associated with *oskar* (*Figure 5D*). This order of events is consistent with association of the degradation factors with intact founder granules leading to *oskar* degradation, followed by the destabilization of founder granules. Furthermore, the percentage of *oskar* particles colocalized with Staufen remained unchanged until nc10, when there was a small decrease in colocalization (*Figure 5E*).

In addition, we analyzed DCP1 association with founder granules containing both *oskar* mRNA and Staufen. Staufen and DCP1 were detected by immunofluorescence together with *oskar* mRNA by FISH and the frequencies of colocalization were quantified. This analysis revealed that $35.0 \pm 5.9\%$ of Staufen particles that colocalized with *oskar* were also colocalized with DCP1 at nuclear cycle 8 (data not shown), consistent with the measurement for *oskar* mRNA (*Figure 4C*). Of the DCP1 particles colocalized with Staufen, $90.0 \pm 5.4\%$ were also colocalized with *oskar* (data not

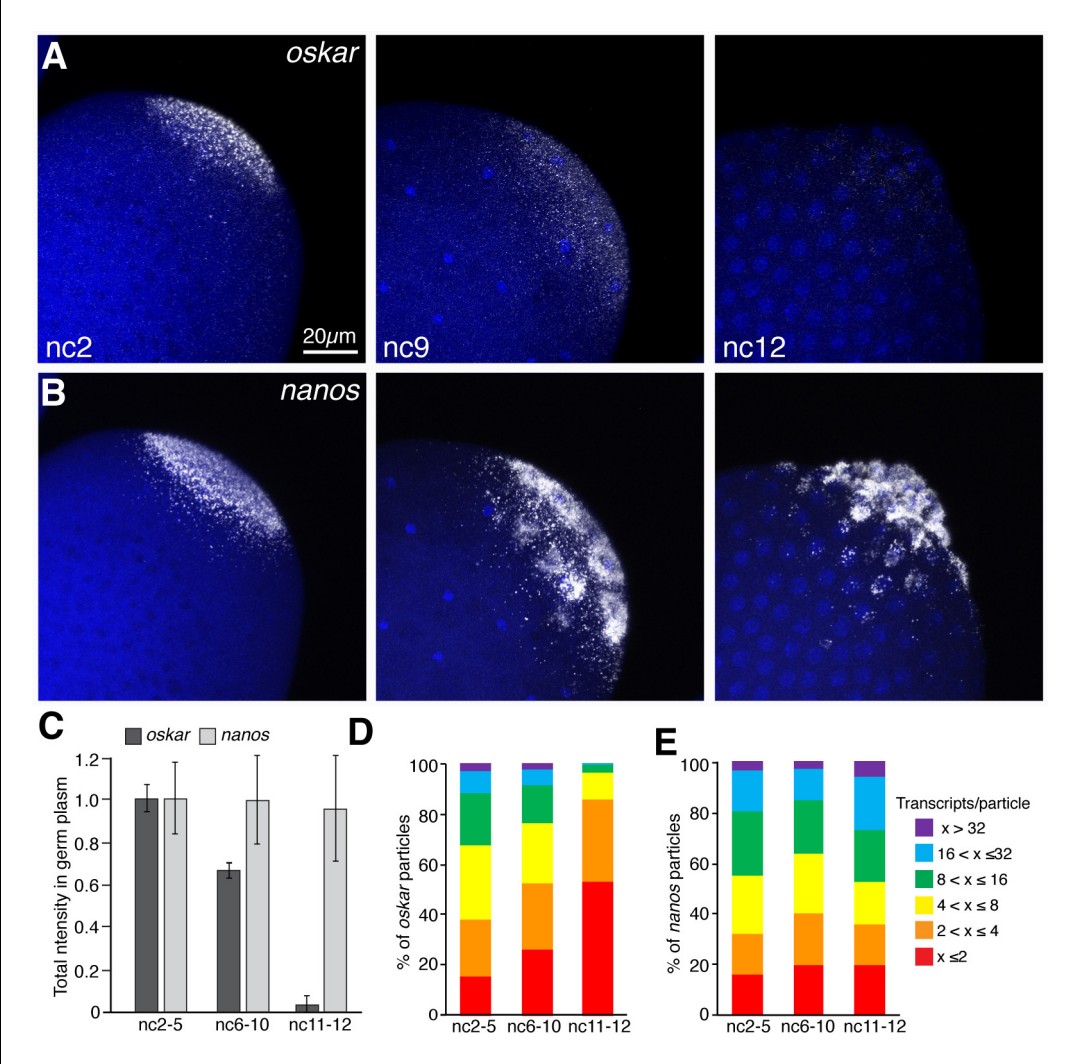

**Figure 3.** *oskar* mRNA is preferentially degraded during pole cell formation. (**A,B**) Confocal z-series maximum intensity projections of the posterior pole of *Drosophila* embryos at nc2, nc9, and nc12, with *oskar* (**A**) or *nanos* (**B**) mRNAs detected by FISH. By nc9, nuclei have arrived at the cortex and by nc12, pole cells have formed. (**C**) Quantification of average total fluorescence intensities of germ plasm localized *oskar* and *nanos* during nc2-5, nc6-10, and nc11-12; n = 7–10 embryos per time period for *oskar*, n = 5 embryos per time period for *nanos*. Values are mean ± s.d., normalized to the values at nc2-5 for each mRNA. (**D,E**) Quantification of the size distributions of founder granules (**D**) and germ granules (**E**) as measured by the number of *oskar* or *nanos* transcripts per particle detected, respectively, from nuclear cycles 2 through 12; (n = 4–20 embryos per time period for germ granules, n = 11–28 embryos per time period for founder granules). Values are mean ± s.d.; ***p<0.001 for *oskar* values at nc6-9 and nc11-12 compared to nc2-5 as determined by Student's t-test.

The online version of this article includes the following source data and figure supplement(s) for figure 3:

**Source data 1.** Related to *Figure 3C*.
**Source data 2.** Related to *Figure 3D,E*.
**Figure supplement 1.** Targeting to germ granules prevents *oskar* degradation.
**Figure supplement 1—source data 1.** Related to *Figure 3—figure supplement 1B*.

shown). Thus, DCP1 associates with intact founder granules. We also analyzed the sizes of *oskar* particles that colocalized with DCP1. If founder granules disassemble prior to *oskar* degradation, DCP1 should preferentially colocalize with small *oskar* particles. However, we found that DCP1 associated with *oskar* particles of all sizes (*Figure 5G,H*) suggesting that *oskar* mRNA is not released from founder granules prior to the onset of degradation.

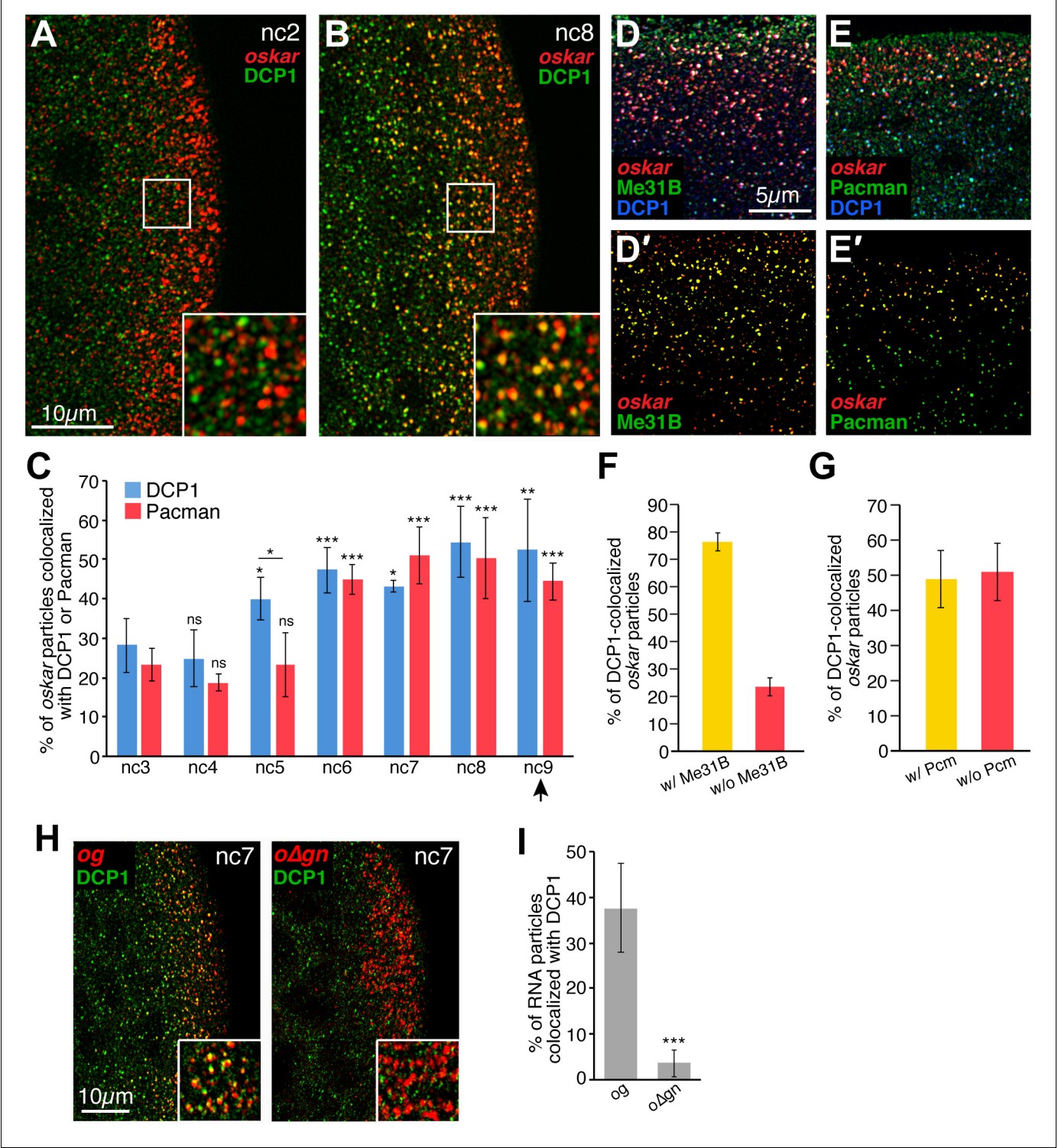

**Figure 4.** Founder granules associate with decapping and degradation factors in the germ plasm in advance of pole cell formation. (**A,B**) Confocal z-series projections of the posterior of nc2 (**A**) and nc8 (**B**) embryos. Anti-DCP1 immunofluorescence (green) was performed together with FISH for *oskar* mRNA (red). Enlargements of the boxed germ plasm regions show colocalization of *oskar* and DCP1 in particles at nc8 but not at nc2. (**C**) Quantification of the proportion of *oskar* particles detected that also contain DCP1 (i.e., are colocalized with DCP1; blue) or Pacman (red) from nc3 to nc9; n = 5–8 embryos for each nc. Arrowhead indicates time of pole cell formation. (**D,D',E,E'**) Confocal z-series projections showing *oskar* mRNA detected by FISH (red) together with DCP1 immunofluorescence (blue) and either direct Me31B-EGFP fluorescence (green; **D,D'**) or Pacman immunofluorescence (green; **E,E'**) in the germ plasm of nc7 embryos. Images in (**D',E'**) are masked by the DCP1 channel so that only the *oskar* and

*Figure 4 continued on next page*

*Figure 4 continued*

either Me31B (D') or Pcm (E') fluorescence signals that overlap with the DCP1 fluorescence signals are visible. (F,G) Quantification of the proportion of *oskar* particles colocalized with DCP1 and Me31B (F) or DCP1 and Pacman (Pcm; G) as compared to the proportion colocalized only with DCP1 in the same images; n = 7 embryos each for Pacman, n = 5 embryos each for Me31B. (H) Confocal z-series projections of the posterior of *og* and *oΔgn* embryos at nc 7. Anti-DCP1 immunofluorescence (green) was performed together with FISH to detect *sfgfp* sequences (red). Enlargements of the boxed germ plasm regions show colocalization of DCP1 with *og* but not *oΔgn* RNA particles in the germ plasm. (I) Quantification of the proportion of *og* and *oΔgn* RNA particles detected that also contain DCP1; n = 8 embryos each. Values are mean ± s.d.; **p<0.01, ***p<0.001 as determined by Student's t-test. In (C) each DCP1 or Pacman is compared to the corresponding value at nc3. DCP1 and Pacman differ significantly at nc5 (p=0.001).

The online version of this article includes the following source data and figure supplement(s) for figure 4:

**Source data 1.** Related to *Figure 4C*.
**Source data 2.** Related to *Figure 4F,G,I*.
**Figure supplement 1.** DCP1 colocalizes with *oskar* but not with germ granule mRNA.
**Figure supplement 1—source data 1.** Related to *Figure 4—figure supplement 1*.

---

Finally, we took advantage of the 5' to 3' directionality of mRNA degradation to monitor the integrity of *oskar* in founder granules before (nc3-4) and after (nc7-8) degradation ensues by performing FISH with differentially labeled probes targeting either the 5' or 3' half of the *oskar* transcript. The ratio of the fluorescence intensities of the 5' and 3' probes (5':3') was then determined for individual *oskar* particles within the germ plasm. For both larger and smaller particles, the 5':3' ratio decreased significantly between nc3-4 and nc7-8, reflecting the loss of sequences from the 5' end of *oskar* (*Figure 5F*). Together, the data presented here support a model in which *oskar* degradation occurs within founder granules.

## *oskar* mRNA degradation in the germ plasm is distinct from degradation of unlocalized *oskar*

Widespread clearance of maternal mRNAs occurs during the MZT and is controlled by both maternal and zygotic degradation factors. The maternal decay machinery is set in motion by egg activation whereas the zygotic pathway is engaged following ZGA (*Laver et al., 2015b*). Genome-wide microarray analysis showed that *oskar* is degraded during the MZT by both maternal and zygotic pathways (*Thomsen et al., 2010*). However, because only 18% of *oskar* mRNA in the embryo is localized to the germ plasm and packaged in founder granules (*Bergsten and Gavis, 1999*; *Little et al., 2015*), this analysis predominantly measured the unlocalized population of *oskar* mRNA in the bulk embryonic cytoplasm. Moreover, previous studies differ in their estimates of the onset of maternal mRNA degradation (*Despic and Neugebauer, 2018*; *Laver et al., 2015b*). Therefore, we sought to determine how degradation of *oskar* in the germ plasm relates to degradation of *oskar* during the MZT.

As a first step, we measured the change in the amount of *oskar* per volume of germ plasm and the change in the amount of *oskar* per volume of bulk cytoplasm in embryos from nc2 to nc12 (*Figure 6A,B*). The first significant decrease in germ plasm localized *oskar* occurred at nc7 (*Figure 6C*), consistent with our analysis of the total amount of localized *oskar* (*Figure 5A*). For unlocalized *oskar*, the first significant decrease occurred at nc11. Consistent with the temporal difference in degradation of the two *oskar* populations, association of unlocalized *oskar* with DCP1 is delayed relative to germ plasm localized *oskar* (*Figure 6D,E*). Together, these results indicate that degradation of *oskar* in the germ plasm begins well in advance of degradation of *oskar* during the MZT.

Next, we asked whether *oskar* degradation in the germ plasm is initiated by the same factors that act later during the MZT. Two RNA-binding proteins, Brain tumor and Pumilio have been shown to direct clearance of *oskar* and numerous other maternal mRNAs in the bulk cytoplasm during the MZT (*Laver et al., 2015a*). The Piwi protein Aubergine has also been implicated (*Barckmann et al., 2015*). We therefore asked whether these proteins also regulate degradation of *oskar* in the germ plasm. In *brain tumor* and *pumilio* mutant embryos, *oskar* was degraded in the germ plasm similarly to wild-type embryos (*Figure 7A,B,D* and data not shown) indicating that Brain tumor and Pumilio do not target *oskar* for degradation in the germ plasm as they do in the bulk cytoplasm. However, *oskar* was partially stabilized in the germ plasm of *aubergine* mutant embryos (*Figure 7A,C,D*). Because most *aubergine* mutant embryos arrest development prior to pole cell formation, we assessed *oskar* degradation in wild-type unfertilized eggs and found that *oskar* was degraded independently of progression through embryogenesis (*Figure 7—figure supplement 1*). Thus, *oskar*

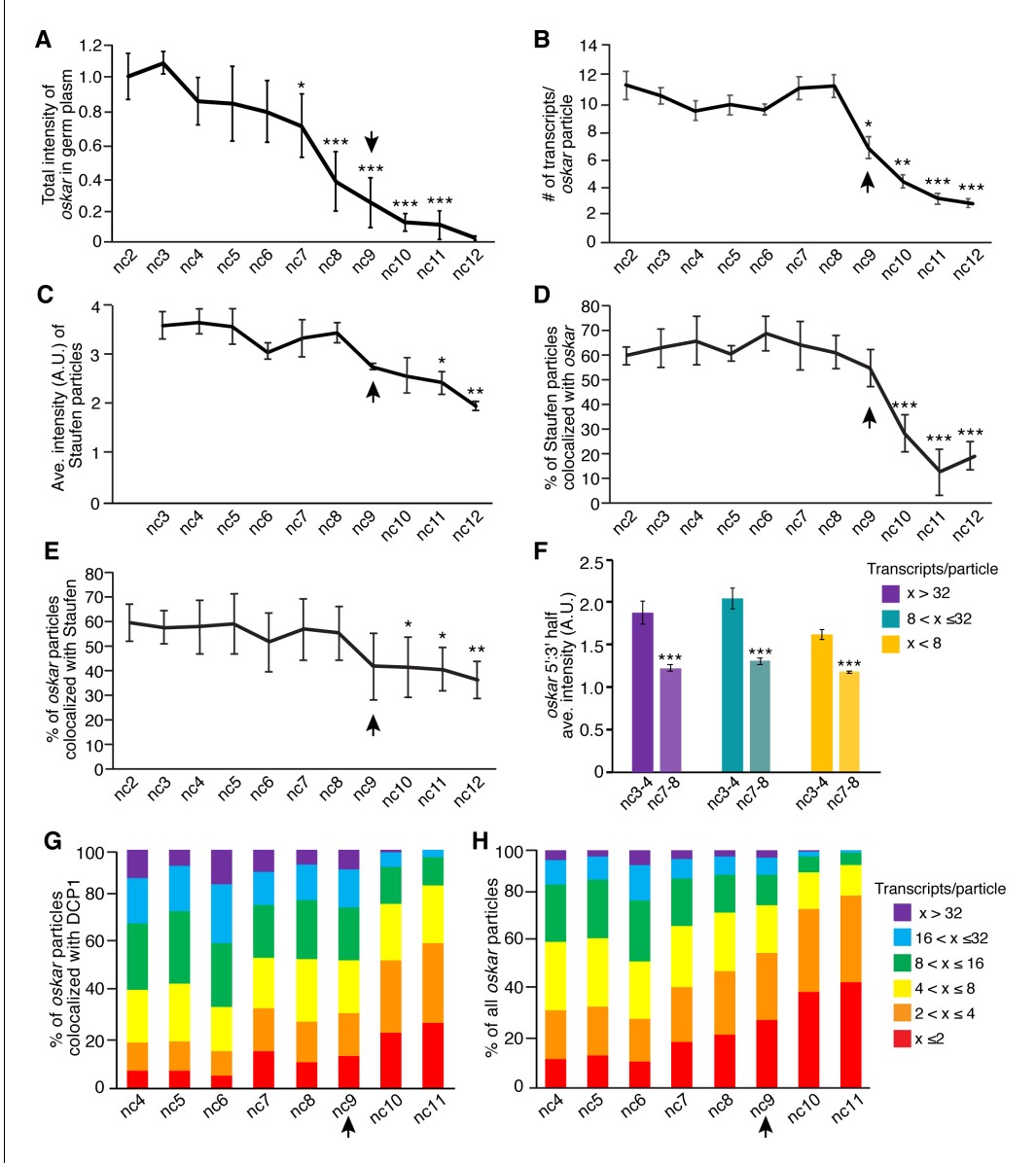

**Figure 5.** Founder granule disassembly follows *oskar* degradation. (**A**) Quantification of total fluorescence intensity of *oskar* in the germ plasm from nc2 to nc12 normalized to the average intensity at nc2; n = 5–13 embryos for each nc. (**B**) Quantification of the average number of mRNAs per detected *oskar* particle from nc2 to nc12; n = 4–10 embryos for each nc. (**C**) Quantification of the average fluorescence intensity (size) of Staufen particles in the germ plasm from nc3 to nc12; n = 3–8 embryos for each nc. (**D**) Quantification of the proportion of detected Staufen particles colocalized with *oskar* from nc2 to nc12; n = 4–10 embryos for each nc. (**E**) Quantification of the proportion of detected *oskar* particles colocalized with Staufen from nc2 to nc12; n = 4–10 embryos for each nc. (**F**) Ratio of the average fluorescence intensities for 5' and 3' *oskar* probes measured particles of the indicated sizes in nc3-4 versus nc7-8 embryos (n = 10 embryos each time period). (**G,H**) Proportion of *oskar* particles in the germ plasm containing the indicated number of mRNAs that are colocalized with DCP1 (**G**) and proportion of all *oskar* particles in the germ plasm with the indicated number of mRNA (**H**); n = 6–9 embryos for each nc. Values in (**A**), (**D**), and (**E**) are mean ± s.d; values in (**B,C,F**) are mean ± s.e.m. Student's t-tests were performed comparing each nuclear cycle to nc2 (**A,B,D,E**), nc3 (**C**), and nc3-4 (**F**); *p<0.05, **p<0.01, ***p<0.001 as determined by Student's t-test. Arrowheads indicates time of pole cell formation.

The online version of this article includes the following source data for figure 5:

**Source data 1.** Related to *Figure 5A,B*.
**Source data 2.** Related to *Figure 5C–E*.
**Source data 3.** Related to *Figure 5F*.
**Source data 4.** Related to *Figure 5G,H*.

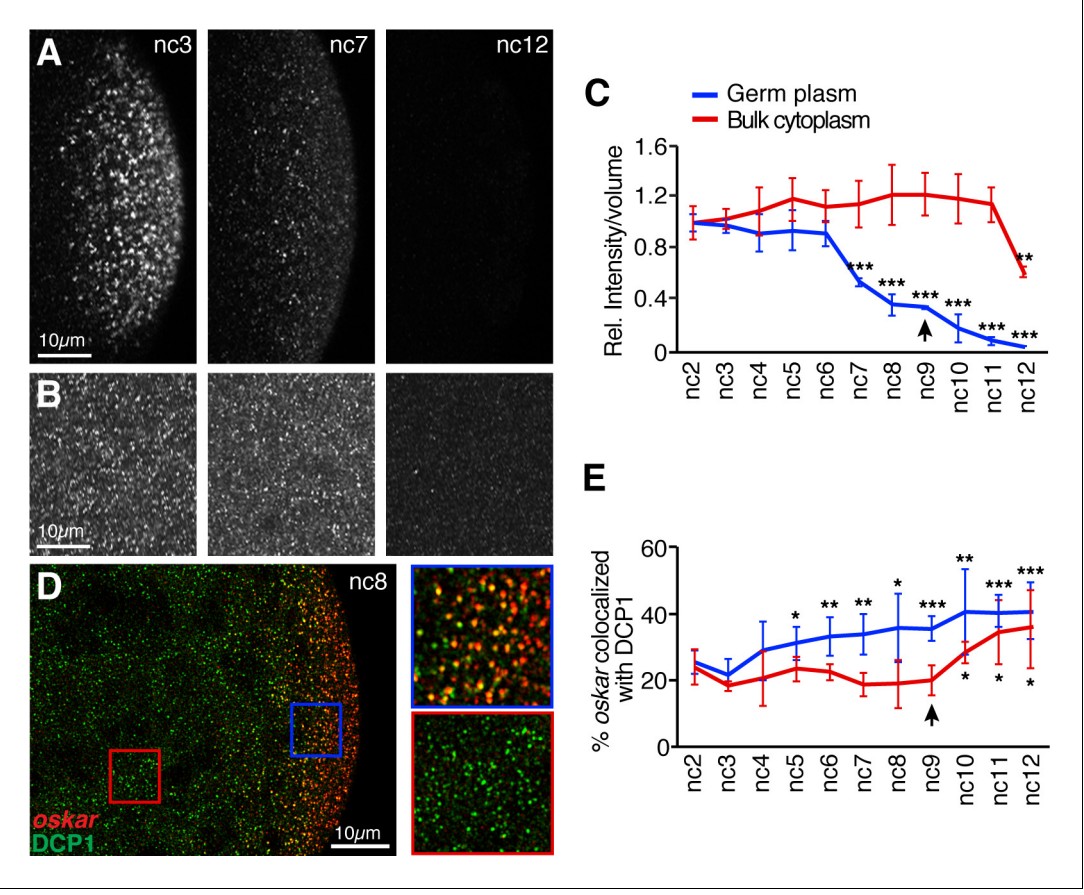

**Figure 6.** *oskar* mRNA is degraded earlier in the germ plasm than in the bulk cytoplasm. (**A,B**) Confocal z-series projections taken at posterior pole (germ plasm, **A**) or the middle region (bulk cytoplasm, **B**) of embryos at nc3, nc7, and nc12 with *oskar* mRNA detected by FISH. Images in (**B**) were captured with higher laser power than images in (**A**) in order to detect small *oskar* RNPs (see Materials and methods). (**C**) Quantification of *oskar* FISH fluorescence intensity per volume in the germ plasm and bulk cytoplasm from nc2 to nc12 normalized to the value at nc2. The first significant changes from nc2 in the germ plasm and bulk cytoplasm occur at nc7 and nc12, respectively; n = 5 embryos for each nc. (**D**) Confocal z-series projection encompassing both the bulk cytoplasm and germ plasm. Anti-DCP1 immunofluorescence (green) was performed together with FISH for *oskar* mRNA (red). Enlargements of the boxed regions show colocalization of *oskar* and DCP1 in the germ plasm but not in the bulk cytoplasm. (**E**) Quantification of the proportion of detected *oskar* particles colocalized with DCP1 in the bulk cytoplasm and in the germ plasm; n = 6–9 embryos for each nc. Values are mean ± s.d. Student's t-tests were performed comparing each nuclear cycle to nc2 for the same region of the embryo; *p<0.05, **p<0.01, ***p<0.001. Asterisks for bulk cytoplasm data points are positioned below the error bars for visibility. Arrowheads indicates time of pole cell formation.

The online version of this article includes the following source data for figure 6:

**Source data 1.** Related to *Figure 6C*.
**Source data 2.** Related to *Figure 6E*.

degradation requires only maternal factors and the stabilization of *oskar* observed in *aubergine* mutant embryos is not simply a consequence of their developmental arrest. Moreover, we found that DCP1 and *oskar* did not colocalize in the germ plasm of *aubergine* mutant embryos (*Figure 7E–G*). Introduction of a functional *gfp-aub* transgene into *aubergine* mutant females rescued both *oskar* degradation and DCP1 colocalization (*Figure 8*). These results suggest a role for Aubergine in recruitment of DCP1 to founder granules, resulting in decapping and subsequent degradation of *oskar*. In addition, *oskar* particle sizes were stabilized in the germ plasm of *aubergine* mutants (*Figure 7H,I*), consistent with *oskar* degradation causing founder granule destabilization.

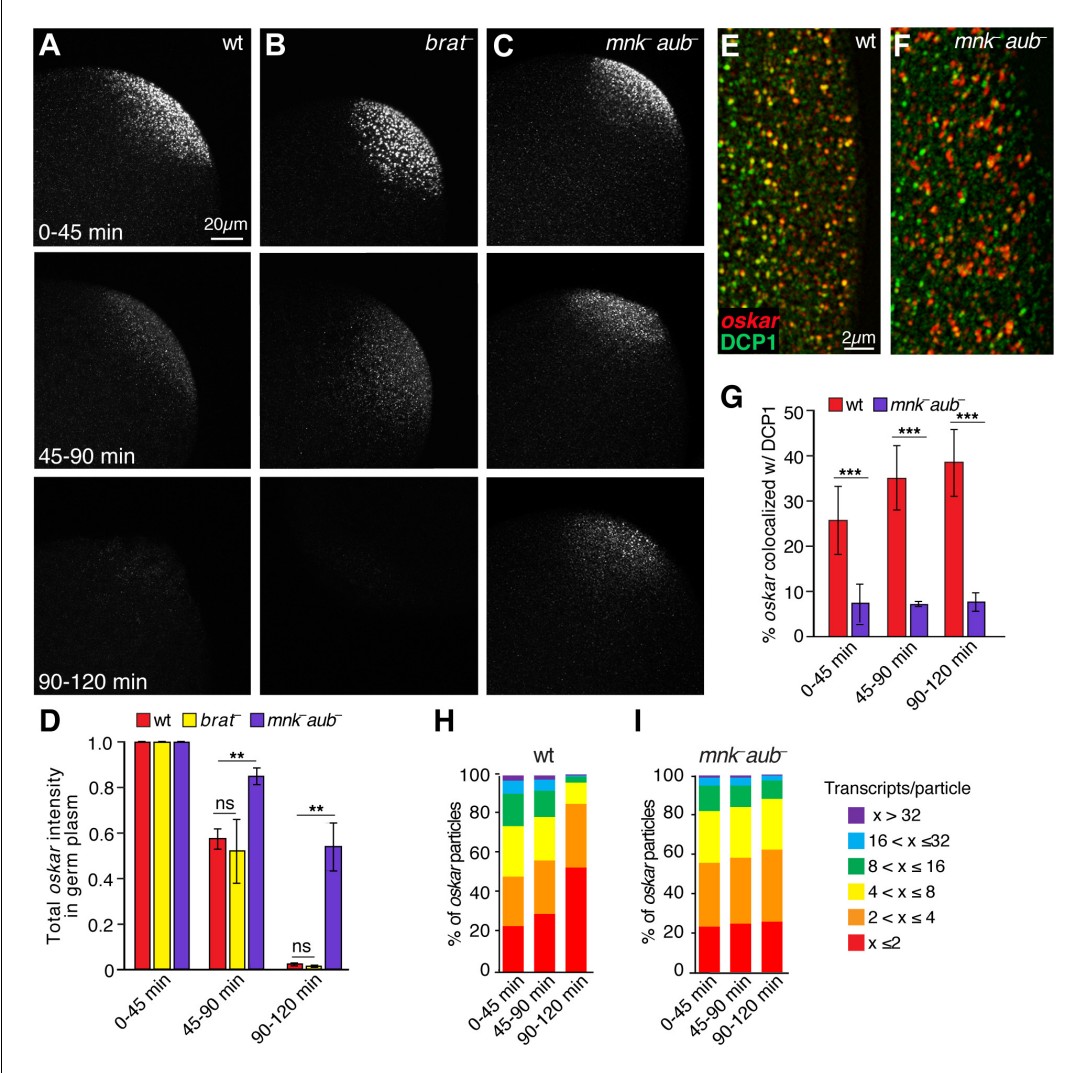

**Figure 7.** Aubergine is required for *oskar* degradation in the germ plasm. (**A–C**) Confocal z-series projections of the germ plasm of wild-type (wt; **A**), *brain tumor* mutant (*brat⁻*; **B**), and *aubergine* mutant (*mnk⁻ aub⁻*; **C**) embryos with *oskar* mRNA detected by FISH. The *mnk* mutation (in *chk2*) bypasses the indirect effect of *aubergine* mutants on *oskar* translation during oogenesis and consequent defects in *oskar* mRNA localization (*Becalska et al., 2011*; *Klattenhoff et al., 2007*); see Materials and methods). (**D**) Quantification of total *oskar* fluorescence intensity in the germ plasm in wt, *mnk⁻ aub⁻* and *brat⁻* embryos. Three biological replicates were performed, each with n = 5–10 embryos per time period for wt; n = 3–7 embryos for *mnk⁻ aub⁻*; n = 3–8 embryos for *brat⁻*. (**E,F**) Confocal z-series projections of a region of the germ plasm in age matched wt (**E**) and *mnk⁻ aub⁻* (**F**) embryos; *oskar* detected by FISH (red) with anti-DCP1 immunofluorescence (green). (**G**) Quantification of colocalization of *oskar* and DCP1 in wt and *mnk⁻ aub⁻* embryos (n = 5–44 embryos each time period). (**H,I**) Quantification of the proportion of detected *oskar* particles with the indicated number of mRNAs per particle number in the germ plasm of wt (**H**) and *mnk⁻ aub⁻* (**I**) embryos; (n = 4–5 embryos each time period for *mnk⁻ aub⁻*, n = 11–28 embryos for wt). Values are mean ± s.e.m., normalized to the value at 0–45 min for each genotype (**D**) and mean ± s.d. (**G**); *p<0.05, **p<0.01, ***p<0.001 as determined by Student's t-test.

The online version of this article includes the following source data and figure supplement(s) for figure 7:

**Source data 1.** Related to *Figure 7D*.
**Source data 2.** Related to *Figure 7G*.
**Source data 3.** Related to *Figure 7H,I*.
**Figure supplement 1.** Comparison of *oskar* stability in fertilized and unfertilized embryos.
**Figure supplement 1—source data 1.** Related to *Figure 7—figure supplement 1*.

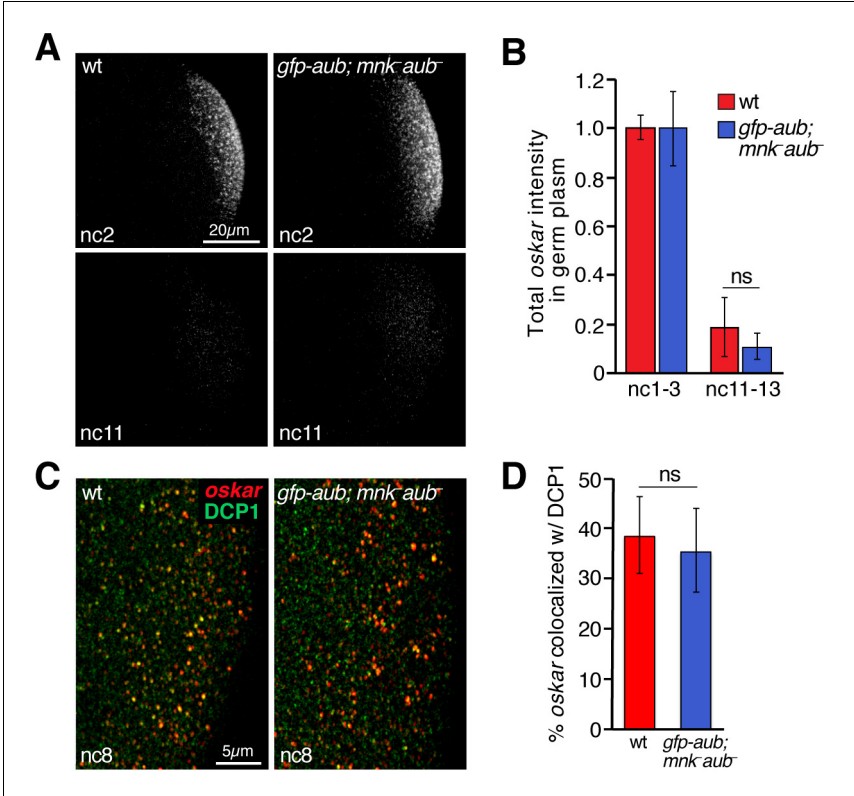

**Figure 8.** Rescue of *oskar* degradation and DCP1 recruitment by *gfp-aub*. (**A**) Confocal z-series projections of the germ plasm of wild-type (wt) embryos and *aubergine* mutant embryos expressing *gfp-aub* (*gfp-aub; mnk⁻ aub⁻*), with *oskar* mRNA detected by FISH. (**B**) Quantification of *oskar* fluorescence intensity in the germ plasm in wt and *gfp-aub; mnk⁻ aub⁻* embryos (n = 7–11 embryos per time period). (**C**) Confocal z-series projections of a region of the germ plasm in age matched wt and *gfp-aub; mnk⁻ aub⁻* embryos; *oskar* detected by FISH (red) with anti-DCP1 immunofluorescence (green). (**D**) Quantification of colocalization of *oskar* and DCP1 in wt and *gfp-aub; mnk⁻ aub⁻* embryos (n = 5 embryos each). Values are mean ± s.d.; ***p<0.001 as determined by Student's t-test.

The online version of this article includes the following source data for figure 8:

**Source data 1.** Related to *Figure 8B,D*.

## Discussion

mRNAs that accumulate in the *Drosophila* germ plasm during oogenesis are partitioned into at least two types of RNPs, germ granules and founder granules. Germ granules contain mRNAs necessary for germline development and promote the inheritance of these mRNAs by pole cells. Founder granules contain *oskar* mRNA, whose translation at the posterior of the oocyte is crucial for germ plasm assembly and germ granule formation but whose accumulation in pole cells is detrimental to germline development. We show that *oskar* RNA itself is toxic to pole cells, necessitating an early-acting mechanism to deplete *oskar* from the germ plasm. Our results indicate that founder granules minimize inheritance of *oskar* by pole cells by recruiting factors for compartmentalized *oskar* degradation.

mRNA degradation is a key feature of early embryonic development, with the majority of the maternal transcriptome turned over and replaced by the zygotic transcriptome during the MZT. Founder granule-associated *oskar* degradation is distinct from degradation of unlocalized *oskar*, *nanos*, and other maternal transcripts during the MZT both temporally and mechanistically. Degradation of *oskar* in the germ plasm occurs well in advance of pole cell formation whereas the majority of MZT-related mRNA degradation occurs after pole cells have formed. Consistent with this timing, *oskar* degradation in the germ plasm is mediated solely by maternal factors whereas *oskar* degradation in the bulk cytoplasm relies on both maternal and zygotic machineries (*Thomsen et al., 2010*). Unlocalized *oskar* and *nanos* are degraded with the same kinetics (*Thomsen et al., 2010*) and

degradation factors (*Laver et al., 2015a*) during the MZT. By contrast, only *oskar* is degraded in the germ plasm and this degradation does not require Brain tumor or Pumilio, but depends in part on Aubergine. Moreover, redirecting *oskar* to germ granules results in its stabilization, akin to *nanos* mRNA. These results, together with evidence that *oskar* degradation precedes founder granule disassembly, indicate that the differential suceptibility of *oskar* and germ granule mRNAs in the germ plasm is imparted by properties of the granules in which they reside.

The selective degradation of *oskar* among the hundreds of different mRNAs that share the germ plasm derives from the exclusive ability of founder granules to recruit decapping and degradation machinery. An outstanding question is what triggers the recruitment of DCP1 to founder granules around nuclear cycle 5. Such a timer could be provided by an as yet unidentified protein whose translation is activated at fertilization and reaches a critical threshold by nuclear cycle 5, similar to temporal control conferred by Smaug during the MZT (*Benoit et al., 2009*). Although DCP1 recruitment requires Aubergine, Aubergine protein is maternally contributed suggesting that it is not the timer (*Kronja et al., 2014*). The requirement for Aubergine in *oskar* degradation is itself surprising. First, Aubergine is a core germ granule protein and has been shown to stabilize germ granule mRNAs through an interaction with the poly(A) polymerase Wispy (*Dufourt et al., 2017*). Second, Aubergine has been proposed to mediate degradation of mRNAs during the MZT by both piRNA-mediated cleavage and recruitment of the RNA-binding protein Smaug and the CCR4-NOT deadenylase complex (*Barckmann et al., 2015*). *oskar* degradation is Smaug-independent, however (*Tadros et al., 2007*). Whether Aubergine might also direct piRNA-mediated degradation of *oskar* in founder granules remains a question for future study but its requirement for association of DCP1 with founder granules suggests that piRNA-mediated degradation may not be the primary mode. Moreover, we did not detect colocalization of *Aubergine* with founder granules suggesting that Aubergine acts indirectly, perhaps by sequestering stability factors. Alternatively, cytoplasmic Aubergine might associate with founder granules only transiently or in small quantities.

Compartmentalization of *oskar* degradation in founder granules may also serve to increase efficacy. The concentration of up to hundreds of *oskar* molecules in founder granules may enhance interaction of *oskar* with the decapping and degradation machinery, increasing reaction rates so that the germ plasm load of *oskar* is sufficiently reduced prior to pole cell formation. Moreover, as non-membrane bound organelles, founder granules would create a hub, bringing the various components of the decapping and degradation machinery into close proximity. Our finding that founder granules are stabilized when *oskar* is stabilized, as in *aubergine* mutants, suggests that *oskar* RNA might regulate the properties of these granules, similarly to the role of RNA in dictating the physical properties of liquid droplets in vitro and in *Ashbya* cells in vivo (*Zhang et al., 2015*).

Co-packaging in RNPs can facilitate the coordinated regulation of cohorts of mRNAs. Germ granules serve such a role by coordinating the delivery of mRNAs to pole cells. Do founder granules contain mRNAs in addition to *oskar* that are toxic to pole cells? Interestingly, under certain conditions, retrotransposon mRNAs can mimic *oskar* and localize to the germ plasm using the same transport machinery (*Tiwari et al., 2017*). Founder granules might also therefore sequester and destroy retrotransposon mRNAs in the germ plasm, preventing their reinsertion into the genome and thus maintaining genome integrity. We suggest that whereas germ granules provide the pole cells with a maternal endowment needed for germline development, founder granules serve an opposing function to protect the germline from inheriting deleterious transcripts.

## Materials and methods

### *Drosophila* strains and genetics

The following *D. melanogaster* alleles and transgenes were used: $y^1$, $w^{67c23}$ (Bloomington *Drosophila* Stock Center [BDSC] 6599) as the wild-type strain; $osk^{54}$ (*Lehmann and Nüsslein-Volhard, 1991*), $osk^{A87}$ (*Vanzo and Ephrussi, 2002*), FRT40 $brat^{11}$ (*Frank et al., 2002*); $mnk^{P6}$, $aub^{HN}$ and $mnk^{P6}$, $aub^{QC}$ (*Klattenhoff et al., 2007*); $pum^{ET3}$ (also called $pum^3$) (*Barker et al., 1992*); $pum^{Msc}$ (*Barker et al., 1992*); *mCherry-vas* (*Lerit and Gavis, 2011*); *GFP-stau* (*Schuldt et al., 1998*); *Me31B-EGFP* (*Buszczak et al., 2007*); *UAS-gfp-aub* (*Harris and Macdonald, 2001*). $brat^{11}$ germline clones were generated by the autosomal FLP-DFS method (*Chou and Perrimon, 1996*). Double mutants for *aub* and *chk2* (*mnk*) were used to bypass the indirect effect of *aubergine* mutants on

*oskar* translation during oogenesis and consequent defects in *oskar* mRNA localization (*Becalska et al., 2011*; *Klattenhoff et al., 2007*). *UAS-gfp-aub* was expressed using the *nos-Gal4-VP16* driver (*Van Doren et al., 1998*) recombined on the same chromosome (gift of P. Macdonald). The *matα-Gal4-VP16* driver (BDSC 7063) was used to express the *UASp-oskΔi1,2-nos3'UTR* (*Little et al., 2015*) and *UASp-osk^{M1,139R}Δi1,2-nos3'UTR* transgenes (this study) in females that also express endogenous, wild-type *oskar*. All genotypes described in the text refer to the maternal genotype, that is the genotype of the mother that generated the embryos analyzed.

## Transgene construction

The pattB-UASp-osk^{M1,139R}Δi1,2-nos3'UTR transgene plasmid was generated from pUASp-oskΔi1,2-nos3'UTR (*Little et al., 2015*) by using site-directed mutagenesis to change the start codons for both Long and Short Oskar protein from ATG to CGC. The UASp-oskΔi1,2-nos3'UTR and UASp-osk^{M1,139R}Δi1,2-nos3'UTR sequences were then inserted into the pattB vector (http://www.flyc31.org/sequences_and_vectors.php). The pattB-osk-sfgfp and pattB-oskΔi1,2-sfgfp-nos3'UTR transgene plasmids are based on the osk::gfp (p2368) construct, which contains an 8 kb *oskar* rescue fragment (*Snee et al., 2007*). The *gfp* sequence fused to the 3' end of *oskar* in osk::gfp was replaced by the *Drosophila* codon optimized sequence encoding superfolder GFP (*sfgfp*) used in the FlyFos phosmid library (*Sarov et al., 2016*). pattB-oskΔi1,2-sfgfp-nos3'UTR also contains sequences encoding a 2xTy1 epitope tag before the *sfgfp* sequence (*Sarov et al., 2016*). All four transgenes were integrated into the attP40 site by phiC31-mediated recombination.

## Embryo collection and fixation

Embryos were collected on apple juice agar plates at room temperature or in a 25°C incubator and staged by nuclear cycle (nc) when possible. When staging by nc was not possible, embryos were collected in a 25°C incubator and staged according to time after egg laying (0–45 min = nc 1–5, 45–90 min = nc 6–10, 90–135 min = nc 11–13). Embryos were dechorionated, fixed, and devitellinized as described in *Abbaszadeh and Gavis (2016)*, then stored in methanol at −20°C for up to 1 month. Embryos used for RT-qPCR were dechorionated, flash frozen in liquid nitrogen, and stored at −80°C.

## Single molecule FISH (smFISH)

smFISH probe sets consisting of 20 nt long oligonucleotides with two nt spacing complementary to *oskar* (CG10901; 99 oligos), *nanos* (CG5637; 63 oligos), *cyclin B* (CG3510; 48 oligos), or *sfgfp* (31 oligos) were designed with the Stellaris Probe Designer (LGC Biosearch Technologies). For *oskar*, *nanos*, and *cyclinB*, oligonucleotides with a 3' amine modification were obtained from Biosearch Technologies, conjugated to Atto 647N or Atto 565 dye (Sigma-Aldrich), then purified by HPLC as previously described (*Raj et al., 2008*). To visualize the 5' and 3' halves of *oskar*, 48 oligos from the complete probe set covering the 5' half of *oskar* were coupled to Atto 565 dye and 48 oligos covering the 3' half were coupled to Atto 647N dye. The *sfgfp* probe set was purchased already conjugated to Quasar 670 dye (Biosearch Technologies). smFISH was performed as described in *Abbaszadeh and Gavis (2016)*. Embryos were mounted under #1.5 glass coverslips (VWR) in Vectashield Mountant (Vector Laboratories) for quantification of total localized fluorescence intensity or Prolong Diamond Antifade Mountant (Thermo Fisher Scientific) for colocalization and particle intensity analyses.

## Immunofluorescence

Embryos were rehydrated stepwise into PBST (PBS, 0.1%Tween-20 [Sigma-Aldrich]) and then blocked in Image it FX (Thermo Fisher Scientific) for 30 min at room temperature followed by washing 3 × 15 min in PBHT (PBS, 0.1% Tween-20, 0.25 mg/ml heparin [Sigma-Aldrich], 50 μg/ml tRNA [Sigma-Aldrich]). Embryos were then incubated in primary antibody diluted in PBHT over night at 4°C with rocking. Primary antibodies: 1:2000 rabbit anti-Staufen (*St Johnston et al., 1991*), 1:1000 mouse anti-DCP1 (*Miyoshi et al., 2009*), 1:500 rabbit anti-Pacman #428 (*Grima et al., 2008*), 1:2000 rabbit anti-Vasa (*Liang et al., 1994*), 1:500 rabbit anti-GFP-Alexa 488 (Invitrogen). Embryos were washed sequentially for 5, 10 and 20 min in PBHT at room temperature, then incubated with the appropriate secondary antibody diluted in PBHT in the dark for 2 hr at room temperature with rocking. Secondary antibodies: 1:1000 goat anti mouse-Alexa 568 (Thermo Fisher Scientific), 1:1000 goat

anti mouse-Alexa 488 (Thermo Fisher Scientific), 1:500 goat anti rabbit-Alexa 568 (Thermo Fisher Scientific). Embryos were washed 3 × 10 min in PBST at room temperature with rocking followed by 2 min incubation in 1.25 µg/ml DAPI in PBST and rinsed four times with PBST. The embryos were then mounted as described above. For double immunofluorescence/smFISH experiments, embryos were refixed in 4% PFA for 30 min at room temperature with rocking and rinsed 4 × with PBST before proceeding with smFISH as described above.

## Quantification of fluorescence intensity

Confocal imaging was performed using a Leica SP5 laser scanning microscope with a 63 × 1.4 NA oil immersion objective and GaAsP 'HyD' detectors in standard detection mode except for the experiments shown in *Figure 3—figure supplement 1*, *Figure 4H,I* and *Figure 8C,D*, which were imaged on a Nikon A1 laser scanning microscope with a 60 × 1.4 NA oil immersion objective and GaAsP detectors. All imaging parameters were kept identical within each experiment. For quantification of total intensity, z-series with a 2 µm step size were used to capture the germ plasm-localized signal. Image processing and analysis were done in FIJI. Z-projections were made with the 'sum slices' function and the threshold adjusted so the entire localized signal was included. The total fluorescence intensity of the localized signal (integrated density function in FIJI) was then measured. The average total intensity for each nc or time period was normalized to the average intensity for the earliest nc or time period in the experiment.

To quantify the fluorescence intensity of RNA within the germ plasm (localized) vs RNA within the bulk cytoplasm (unlocalized), two images were captured for each embryo: one at lower laser power optimized for the brighter germ plasm RNA signal and one at higher laser power optimized for the unlocalized RNA signal. Image processing and analysis were done in FIJI. Z-projections spanning 12 µm (6 µm above and below the mid-point of the germ plasm in the z-axis) were made with the 'sum slices' function. A ROI of approximately 1600 µm$^3$ within the germ plasm and a ROI of approximately 83100 µm$^3$ within bulk cytoplasm was selected for each embryo. A large ROI was chosen for the bulk cytoplasm to miminize variation in signal due to interference from yolk. The integrated density of the ROI region was divided by the corresponding volume.

## Quantification of particle size and colocalization

Confocal imaging was performed using a Leica SP5 laser scanning microscope with a 63 × 1.4 NA oil immersion objective and GaAsP 'HyD' detectors in photon counting mode except for the experiment shown in *Figure 3H,I*, which was imaged on a Nikon A1 laser scanning microscope with a 60 × 1.4 NA oil immersion objective and GaAsP detectors. An optical zoom of 1.52× (SP5) or 2.9× (A1) was applied to achieve a pixel size of 72 × 72 nm. An approximate 5 µm z-series was taken of the posterior third of the embryo with a step size of 340 nm. The fluorescence intensities of individual particles of *oskar*, *nanos*, and Staufen and were quantified as previously described (*Niepielko et al., 2018*), with unlocalized *nanos* and *oskar* particles assumed to contain an average of 1 and 2 transcripts, respectively, as previously determined (*Little et al., 2015*). With these assumptions, the average intensities of unlocalized particles were used to determine the number of transcripts in germ plasm localized particles (particle size) of the same image. In addition to allowing the determination of the number of transcripts in RNP granules, this method internally normalizes each image allowing direct comparisons. Internally normalized data points for embryos obtained from up to four independent experiments were pooled to obtain sufficient sample sizes; data are reported as the average of the average particle size for each embryo (mean) ± standard error of the mean. Colocalization between *oskar* and DCP1, Me31B, Pacman, or Staufen and between *cyclin B* and DCP1 was measured as previously described (*Niepielko et al., 2018*). Data points for embryos from up to four independent experiments were pooled to obtain sufficient sample sizes.

## Live imaging and particle tracking

Live imaging of founder granules and germ granules was performed on a Nikon Ti-E with Yokogawa spinning disc module (CSU-21) with a 60 × 1.4 NA oil immersion objective. Embryos were dechorionated and mounted in halocarbon oil 95 (Halocarbon Products Corporation) under a #1.5 glass coverslip (VWR) on a Lumox dish (Sarstedt), oriented with the dorsal side closest to the coverslip.

Movies were acquired at 100 ms exposure in two channels with a frame rate of 5 frames per second. Particles were tracked using the Autoregressive Motion algorithm of Imaris version 9.1.2 (Bitplane).

## Immunohistochemistry

Fixed embryos were rehydrated stepwise into PBST, then blocked by incubation in BBT (PBST supplemented with 0.1% globulin free BSA [Sigma-Aldrich]), 5 × 25 min at room temperature with rocking. The embryos were then incubated with 1:2000 rabbit anti Vasa antibody (*Liu et al., 2003*) diluted in BBT overnight at 4°C with rocking, followed by 10, 20 and 30 min washes with BBT at room temperature with rocking. The embryos were blocked again for 30 min with BBT + 2% normal goat serum (NGS), then incubated with 1:2000 biotin goat anti rabbit antibody (Jackson Immuno Research Laboratories Inc) in BBT + 2% NGS for 2 hr at room temperature with rocking. The embryos were then washed for 10, 20, and 2 × 30 min in PBSTx (PBS, 0.1% TritonX100 [Sigma-Aldrich]) at room temperature with rocking. Secondary antibody amplification was performed with Vectastain reagent according to the manufacturer's instructions (Vectastain Elite PK6100 ABC kit; Vector Labs) and detected using peroxidase immunohistochemistry. Embryos were mounted in 80% glycerol and imaged using Nomarski optics.

## Immunoblotting

Ovaries were dissected from female fed with yeast paste for 2 days at 25°C, then frozen in liquid $N_2$. Frozen ovaries were homogenized in boiling 2 × sample buffer (0.125 M Tris-HCl pH 6.8, 4% SDS, 20% glycerol, 5 M urea, 0.1 M DTT, 0.01% bromophenol blue) and boiled for 5 min. Samples were spun for 15 min at 4°C and the supernatant was run on a 10% SDS-PAGE gel. The membrane was blocked 3 × 10 min in Blotto (10 mM Tris-HCl pH 7.5, 150 mM NaCl, 0.1% Tween-20, 5% nonfat dry milk). The membrane was then cut just below the 100 kDa marker; the top portion was incubated at 4°C overnight in Blotto with 1:10,000 rabbit anti-Kinesin heavy chain (Cytoskeleton Inc) and the bottom portion was incubated similarly with 1:2000 rabbit anti-Oskar (gift of A. Ephrussi). After washing 3 × 15 min in Blotto at room temperature, the membranes were incubated in 1:2000 HRP donkey anti rabbit antibody for 1.5 hr at room temperature. The membrane was washed 3 × 15 min in TBST (10 mM Tris-HCl pH 7.5, 150 mM NaCl, 0.1% Tween-20). Antibodies were detected using the Lumi-Light Plus Western Blotting Substrate (Roche) and the blot was imaged using an iBright FL1000 Imaging System (Invitrogen).

## RT-qPCR

RNA was extracted from dechorionated embryos using the RNeasy kit (Qiagen). 1 µg total RNA was used to generate cDNA using the Quantitect RT kit (Qiagen). 2 µl cDNA was combined with 25 µl 2 × TaqMan Gene Expression Master Mix (Thermo Fisher Scientific), 2.5 µl of 20 × TaqMan Gene Expression Assay (Thermo Fisher Scientific, *oskar* custom – Dm02134535, 4351372 or *RpL7* Dm01817653, 4351372), and 20.5 µl of nuclease free $H_2O$. qPCR was performed on an Applied Biosystems 7900HT standard 96-well qPCR instrument. Three biological replicates were performed with three technical replicates each, all using a CT threshold of 0.6613619. Technical replicates were averaged and copy number determined using previously established standard curves (*Eagle et al., 2018*). Copy number of the three biological replicates was normalized to the *RpL7* control and shown as mean ± s.d.

## Acknowledgements

We are grateful to P Lasko, S Newbury, H Siomi, and D St Johnston for antibodies, P Macdonald for plasmid DNA, and P Macdonald, W Theurkauf, and E Wieschaus for fly stocks. We thank G Laevsky for assistance with confocal imaging. S Chatterjee for technical assistance, and W Eagle, F Hughson, and M Niepielko for critical comments on the manuscript. This work was supported by National Institute of Health (NIH) grants R01 GM067758 and R35 GM126967 to ERG. CEE was supported by NIH training grant T32 GM007388.

## Additional information

### Funding

| Funder | Grant reference number | Author |
| --- | --- | --- |
| National Institutes of Health | R01 GM067758 | Elizabeth R Gavis |
| National Institutes of Health | R35 GM126967 | Elizabeth R Gavis |
| National Institutes of Health | T32 GM007388 | Elizabeth R Gavis |

The funders had no role in study design, data collection and interpretation, or the decision to submit the work for publication.

### Author contributions

Catherine E Eichler, Conceptualization, Formal analysis, Investigation, Methodology; Anna C Hakes, Brooke Hull, Formal analysis, Investigation; Elizabeth R Gavis, Supervision, Funding acquisition, Investigation, Project administration

### Author ORCIDs

Elizabeth R Gavis (iD) https://orcid.org/0000-0003-0251-0760

### Decision letter and Author response

Decision letter https://doi.org/10.7554/eLife.49988.sa1
Author response https://doi.org/10.7554/eLife.49988.sa2

## Additional files

### Supplementary files

• Transparent reporting form

### Data availability

Source data files have been provided for all figures and figure supplements.

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
