## [Decision Letter]

**Acceptance summary:**

The manuscript provides novel insights into the regulation of *oskar* mRNA degradation. The authors show that *oskar* mRNA is partitioned into specific granules, called founder granules. These founder granules serve to compartmentalize the degradation of *oskar* mRNA during embryogenesis, thus minimizing its inheritance by pole cells, where it is toxic. The authors show that this compartmentalized degradation is distinct from the mechanisms that regulate the degradation of *oskar* and other mRNAs during the maternal to zygotic transition.

**Decision letter after peer review:**

Thank you for submitting your article "Compartmentalized *oskar* degradation in the germ plasm safeguards germline development" for consideration by *eLife*. Your article has been reviewed by three peer reviewers, including Michael Buszczak as the Reviewing Editor and Reviewer #1, and the evaluation has been overseen by Michael Eisen as the Senior Editor. The following individual involved in review of your submission has agreed to reveal their identity: Francisco Pelegri (Reviewer #3).

The reviewers have discussed the reviews with one another and the Reviewing Editor has drafted this decision to help you prepare a revised submission.

Summary:

In this paper, Ruesch and Gavis describe two types of granules that are vital for germ plasm formation and germ cell development: germ granules and founder granules. During oogenesis, the protein *Oskar* initiates the formation of germ granules through the recruitment of various other core components. As oogenesis proceeds, various mRNAs, including nanos, continued to be recruited to germ granules. By contrast, *oskar* mRNA is not, and instead populates a distinct type of granule that the authors refer to as founder granules. Work presented here shows that the separation of various mRNAs into these two distinct granules is important for the proper establishment of germ plasm. Ectopic recruitment of *oskar* mRNA to germ granules results in a decrease in the number of pole cells during embryogenesis. Further work shows that this likely depends on *oskar* mRNA and not on any ectopic *Oskar* protein. The founder granules exhibit limited mobility and do not disperse during pole cell formation. Germ plasm localized *oskar* mRNA is degraded during pole cell formation, and correlative data suggests these transcripts are degraded with founder granules. This degradation depends on decapping enzymes and appears distinct from the degradation machinery that functions during the MZT. Interestingly, founder cell degradation of *oskar* mRNA does not occur in *aub* mutants. Consistent with this observation, *aub* mutants fail to recruit Dcp1 to founder granules.

The concept that there are multiple types of granules undergoing different fates fits well with and contributes new facets to the growing understanding of fluid phase compartmentalization, and as such the studies presented constitute an important first example of such processes. The properties of osk RNA, essential to generate the germ plasm, appear to be detrimental at later stages, perhaps through interfering with another related biophysical pathway. Thus osk RNA needs to be removed in a timely manner. This work will be of broad interest to the field.

However, the manuscript has a number of issues that should be addressed before publication.

Essential revisions:

1) Further characterization of founder granules and *oskar* mRNA localization to these structures is warranted. Is recruitment of the degradation machinery osk-dependent? Does the oΔn mRNA localized to the germ granules lack localization with these factors? Are there specific elements within *oskar* mRNA that drive its temporally specific degradation or is recruitment to founder granules sufficient to cause other mRNAs to be degraded. Do the authors have insights into what regions of *oskar* mRNA are necessary for its toxic effects? Osk transcript degradation in mutants or knockdowns of *Dcp1, Me31B* and *pacman* should also be analyzed, if possible.

2) The role of Aubergine (and potentially piRNAs) in the degradation of osk, presented in Figure 6 is interesting, but not convincing and also lacks experimental support. First, the work cited implicating AUB in the degradation of osk in the bulk cytoplasm is itself not entirely convincing [Barckmann et al., 2015]. Second, and more concerning, the observations mentioned in the Discussion section that AUB localizes to germ granules and not founder granules undermines the data presented in this manuscript that a) DCP1 (and later ME31B and PCM) colocalize to osk founder granules associated with osk degradation; and (b) oΔn mRNA localized to germ granules (which contain AUB) does not undergo degradation. Therefore, the conclusion that AUB is involved in regulating osk degradation is not concrete, and the authors cannot conclude that AUB plays a role in the recruitment of DCP1 to osk in founder granules based on the data presented here. The authors should experimentally address these concerns or consider deleting the Aub section altogether.

---

## [Author Response]

Essential revisions:1) Further characterization of founder granules and oskar mRNA localization to these structures is warranted. Is recruitment of the degradation machinery osk-dependent? Does the oΔn mRNA localized to the germ granules lack localization with these factors? Are there specific elements within oskar mRNA that drive its temporally specific degradation or is recruitment to founder granules sufficient to cause other mRNAs to be degraded. Do the authors have insights into what regions of oskar mRNA are necessary for its toxic effects? Osk transcript degradation in mutants or knockdowns of Dcp1, Me31B and pacman should also be analyzed, if possible.

a) Founder granules do not exist in the absence of *oskar* mRNA so we cannot directly test whether it is *oskar* RNA that is recruiting the machinery to founder granules. However, we have now shown experimentally that *oΔn* RNA in germ granules does not colocalize with DCP1, indicating that *oskar* RNA per se is not sufficient to recruit degradation machinery. We have added these data (Figure 4H,I.)

b) Currently, *oskar* is the only known founder granule resident RNA. We previously performed FISH for numerous transcripts that are enriched in the germ plasm but thus far have not identified any other RNA localized to founder granules. *oskar* localization is unusual in requiring splicing of the first intron to create a stem-loop localization element that acts in an unknown manner with the exon junction complex that is deposited during this splicing event. This localization signal mediates transport of *oskar* to the posterior during stages 7-10 of oogenesis. A second wave of *oskar* localization occurs at during stages 11-13, and it is not known what sequences in *oskar* mediate this localization. The bottom line is that we do not know how to drive RNAs into founder granules so we are not able to test whether recruitment to founder granules is sufficient to cause other mRNAs to be degraded.

c) Related to the point above, *oskar* is a very complicated RNA, with numerous elements that regulate various aspects of its transport, anchoring, and translational regulation scattered throughout the transcript. Dissection of the regions that control the degradation would take several years of work and is complicated by the need to avoid compromising these other functions – it is thus beyond the scope of this paper.

d) We do not know what regions of *oskar* are necessary for the toxic effects. C.E.E. (first author)'s thesis committee discouraged her from pursuing this question because they felt that she was unlikely to learn anything relevant to "normal" *oskar* function or germ cell development. We agreed that it was better to stay focused on upstream events (e.g., regulation of *oskar*) given limits of time and resources.

e) Unfortunately, is not possible to analyze *oskar* degradation in *Dcp1*, *Me31B* and *pacman* mutants because oogenesis is arrested. *Dcp1* mutants arrest at stage 6, Me31B mutant egg chambers degenerate at or before stage 10, and pacman mutant egg chambers degenerate after stage 9. We tried a partial knockdown of *pacman* by RNAi and although we observed a small effect, the data were highly variable and not statistically significant (likely due to the partial nature of the knockdown).

2) The role of Aubergine (and potentially piRNAs) in the degradation of osk, presented in Figure 6 is interesting, but not convincing and also lacks experimental support. First, the work cited implicating AUB in the degradation of osk in the bulk cytoplasm is itself not entirely convincing [Barckmann et al., 2015]. Second, and more concerning, the observations mentioned in the Discussion section that AUB localizes to germ granules and not founder granules undermines the data presented in this manuscript that a) DCP1 (and later ME31B and PCM) colocalize to osk founder granules associated with osk degradation; and (b) oΔn mRNA localized to germ granules (which contain AUB) does not undergo degradation. Therefore, the conclusion that AUB is involved in regulating osk degradation is not concrete, and the authors cannot conclude that AUB plays a role in the recruitment of DCP1 to osk in founder granules based on the data presented here. The authors should experimentally address these concerns or consider deleting the Aub section altogether.

We agree that this is a surpising result. We have now performed a rescue experiment by expressing a functional *gfp-aub* transgene in *mnk^–^ aub^–^*females. We show that degradation of *oskar* is indeed restored by expression of *gfp-aub*. Moreover, we show that DCP1 is recruited to *oskar* in these embryos. We have added a figure (Figure 8) with these new data.

I had hoped to be able to test whether the unlocalized population of Aubergine that is not in germ granules is the relevant source by expressing GFP-Aub mutants that either cannot be methylated or cannot bind piRNAs and consequently cannot associate with germ granules (obtained from the Aravin lab). Unfortunately, these proteins appear to be toxic and *mnk^–^ aub^–^*females expressing them produce few eggs, all of which disintegrate rapidly after laying.

We have also toned down our conclusions in the Discussion. In particular, we say no longer state that Aubergine recruits DCP1 to founder granules because it implies a direct role for Aubergine. Rather we have changed the wording to state that recruitment of DCP1 to founder granules requires Aubergine and that Aubergine is required for *oskar* degradation. We suggest that the cytoplasmic pool of Aubergine could be the source and/or that Aubergine may act indirectly. Finally, we revised our citation of the previous work to say "Aubergine has been proposed to mediate degradation of mRNAs during the MZT…." which we also hope addresses the reviewer's concerns.